# Implementation and Testing of V2I Communication Strategies for Emergency Vehicle Priority and Pedestrian Safety in Urban Environments

**DOI:** 10.3390/s25020485

**Published:** 2025-01-16

**Authors:** Federica Oliva, Enrico Landolfi, Giovanni Salzillo, Alfredo Massa, Simone Mario D’Onghia, Alfredo Troiano

**Affiliations:** 1Netcom Engineering S.p.A., Via Nuova Poggioreale, Centro Polifunzionale, Tower 7, 5th Floor, 80143 Naples, Italy; f.oliva@cons.netcomgroup.eu (F.O.); g.salzillo@netcomgroup.eu (G.S.); s.donghia@netcomgroup.eu (S.M.D.); a.troiano@netcomgroup.eu (A.T.); 2“Ettore Pancini” Physics Department, University of Naples Federico II, Monte Sant’Angelo Campus, Via Cintia 21, 80126 Naples, Italy; alfredo.massa@unina.it

**Keywords:** vehicular communication, V2I communications, intelligent system transportation, OBU, RSU, Dedicated Short-Range Communications, priority intersection management, pedestrian in signalized crosswalk warning

## Abstract

This paper explores the development and testing of two Internet of Things (IoT) applications designed to leverage Vehicle-to-Infrastructure (V2I) communication for managing intelligent intersections. The first scenario focuses on enabling the rapid and safe passage of emergency vehicles through intersections by notifying approaching drivers via a mobile application. The second scenario enhances pedestrian safety by alerting drivers, through the same application, about the presence of pedestrians detected at crosswalks by a traffic sensor equipped with neural network capabilities. Both scenarios were tested at two distinct intelligent intersections in Lioni, Avellino, Italy, and demonstrated notable effectiveness. Results show a significant reduction in emergency vehicle response times and a measurable increase in driver awareness of pedestrians at crossings. The findings underscore the potential of V2I technologies to improve traffic flow, reduce risks for vulnerable road users, and contribute to the advancement of safer and smarter urban transportation systems.

## 1. Introduction

Road accidents and traffic safety are critical issues faced by modern societies, with millions of lives affected each year. The need to improve safety for all road users, including vehicles, emergency responders, and pedestrians, has led to the development of advanced technologies and strategies aimed at reducing accidents and mitigating their consequences. Emergency vehicles, such as ambulances and fire trucks, play a vital role in providing timely assistance, but their swift movement through traffic poses additional challenges to road safety. In fact, their urgent need to reach destinations quickly can increase the risk of collisions, especially at intersections or in congested urban areas. It is essential to understand the statistics and contributing factors behind emergency vehicle accidents in order to enhance road safety. Accidents involving emergency vehicles can occur with various types of vehicles; however, the most frequently involved are ambulances, fire trucks, and police cars. Among these, ambulance accidents represent the largest proportion of emergency vehicle collisions, followed by incidents involving fire trucks and police vehicles. The high speeds at which ambulances travel to respond to emergencies contribute significantly to the frequency of these accidents [1].

In 2022, a total of 224 people lost their lives in crashes involving emergency vehicles in the United States. Of these fatalities, 50% were occupants of non-emergency vehicles, while 22% were pedestrians. Moreover, ambulance accidents can pose serious risks, as they often involve patients who are already injured or unwell. Indeed, an ambulance can be involved in a traffic accident even after departing from the emergency scene, as it works to reach the destination of care as swiftly as possible. Emergency vehicle drivers accounted for 16% of the deaths, and passengers in emergency vehicles made up approximately 8%. In addition, the same study from which the previous data were derived focused on examining the potential increase in accidents when ambulances use or do not use lights and sirens, distinguishing between the phase in which the ambulance travels to the scene of assistance and the phase in which the patient is transported to the hospital. During the first phase, it has been shown that when an ambulance does not use lights and sirens, the number of incidents is 4.6 per 100,000 calls, and it rises to 5.5 when lights and sirens are activated. Conversely, during the patient transport phase, the results of this study indicate that the risk of accidents without lights and sirens is 7.0 per 100,000 transports, increasing to 16.5 when lights and sirens are in use [2]. Fire trucks are generally larger and heavier than other emergency vehicles, making them more challenging to navigate through traffic. In December 2024, the National Fire Protection Association (NFPA) released a report detailing firefighter injuries in the United States. As reported in [3], in 2023, there were 19,225 incidents involving fire department emergency vehicles while returning from various calls. This figure highlights the frequency of collisions faced by emergency responders during their operations. These collisions resulted in nearly 1450 injuries, representing about 17% of all firefighter injuries for that year. This statistic underscores the considerable dangers firefighters encounter not only during emergency responses but also on their return trips. Tragically, 14 firefighters either lost their lives or experienced symptoms while responding to or returning from incidents, with 8 fatalities occurring during responses and 6 while returning from calls [4]. Accidents involving emergency vehicles can arise from a wide range of causes, often stemming from a combination of factors. In order to reach emergency scenes quickly, they may need to run red lights or disregard stop signs, which increases the likelihood of collisions with other vehicles or pedestrians. Additionally, other drivers may not be aware of an approaching emergency vehicle, particularly in areas with limited visibility, leading to sudden lane changes or a failure to yield, which can result in accidents. Other contributing factors include driver distractions, such as radio communication or navigation systems [1,5], which can divert attention away from the road, as well as road conditions [6,7] that may hinder safe operation.

Pedestrians rank among the most vulnerable groups of road users and can experience accidents both during emergency situations and in everyday urban settings. The urgent responses of emergency vehicles can create dangerous conditions for pedestrians. However, pedestrian accidents also occur in the absence of emergency circumstances. In 2021 alone, NHTSA reported over 7000 pedestrian fatalities in the United States [8,9], marking the highest total since 1981. Moreover, the World Health Organization (WHO) estimates that approximately 1.3 million people die annually due to road traffic accidents worldwide [10]. According to a study on identifying high-risk areas for vehicle–pedestrian collisions using Artificial Neural Networks (ANNs) referenced in [11], signalized intersections frequently report significantly higher rates of fatal pedestrian accidents compared with other locations. This heightened risk can be attributed to a combination of factors, such as dense pedestrian traffic, insufficient or poorly designed pedestrian infrastructure, illegal crossing behaviors, and frequent signal violations [12]. A significant concern is the alarming number of fatalities among school-age children (18 and younger) related to transportation incidents. Between 2005 and 2014, of the 304 school-age fatalities, most occurred during critical hours: from 7:00 to 8:00 a.m. and from 3:00 to 4:00 p.m. [13]. These data underscore the urgent need for interventions to safeguard pedestrians, particularly near schools, where children are at a heightened risk during these peak hours.

Given the issues presented, it is evident that developing technological solutions to reduce the high incidence of accidents and fatalities is essential. Numerous investments have been made to support research activities dedicated to developing new technologies for vehicle-to-environment communication. Indeed, a research program called Borgo 4.0 has been promoted by ANFIA (Associazione Nazionale Filiera Industria Automobilistica) with the involvement of a public–private partnership consisting of 54 companies, 3 public research centers, 5 Campanian universities, and the CNR (Consiglio Nazionale delle Ricerche). The Borgo 4.0 platform accommodates the C-Mobility project to create new intelligent solutions dedicated to vehicle-to-vehicle communications and the development of safety applications. This paper aims to showcase the development and testing process of two IoT applications designed to facilitate Vehicle-to-Everything (V2X) communication, specifically between vehicles and two distinct intelligent intersections established in Lioni, Campania, Italy. The first application enhances the passage of emergency vehicles by sending alert notifications to approaching drivers via a mobile application as they approach the intelligent intersection. The second scenario specifically focuses on the safety of pedestrians crossing near a school. It utilizes the same application to alert drivers approaching the intelligent intersection about the presence of a pedestrian detected by a traffic sensor who is currently in the process of crossing. This feature aims to enhance safety for vulnerable road users, particularly children, by ensuring that drivers are informed and can take necessary precautions in school zones where pedestrian activity is typically high.

## 2. State of the Art

The era in which we live is characterized by a new way of moving from one place to another, with the ultimate goal of making these movements more efficient and less polluting. The rapid advancement of technology is transforming urban environments into smart cities, where digital infrastructure and interconnected systems aim to improve the quality of life for citizens. A network of connected vehicles and roadside access points, also known as Vehicular Ad Hoc Networks (VANETs) [14], has been identified as a key technology to enable intelligent transportation systems (ITSs) designed for real-time Vehicle-to-Vehicle (V2V) and Vehicle-to-Infrastructure (V2I) communications based on a Dedicated Short-Range Communications (DSRC) protocol operating in the 5.9 GHz frequency band. A significant aspect of DSRC is its implementation of the IEEE 802.11p standard, which is an extension of the well-known Wi-Fi protocols tailored specifically for vehicular environments [15,16]. IEEE 802.11p enables devices to communicate directly without relying on a cellular network, ensuring a stable connection even in high-density traffic scenarios. Each communication unit, whether installed on a vehicle or situated on a roadside, is designated as an ITS station [17]. ITS stations are primarily divided into two categories: On-Board Units (OBUs) and Road-Side Units (RSUs). OBUs are devices installed on vehicles, while RSUs are placed along roadways or at critical infrastructure points such as intersections and traffic signals. To enable communication, these units must adhere to the same protocol, which typically depends on the geographical area. For example, the European standard for vehicular communication is defined by the European Telecommunications Standards Institute (ETSI). The latter defines a set of 16 most relevant and interesting use cases for the automotive industry [18], which are for the road safety, as follows:1.Road work warning;2.Stop sign violation;3.Traffic jam ahead warning;4.Car breakdown warning;5.Slow vehicle warning;6.Approaching emergency vehicle.
For traffic efficiency management:7.In-vehicle signage;8.Regulatory and contextual speed limits;9.Traffic info and recommended itinerary;10.Limited access warning;11.Decentralized floating car data;12.Green light optimal speed advisory.
Other:13.Vehicle software provisioning and update;14.Fleet management;15.Local electronic commerce;16.Insurance and financial services.

V2X communications, combined with the incorporation of sensors in vehicles that enable them to collect information from their environment, facilitate the transformation of standard vehicles into connected autonomous vehicles (CAVs) [19]. In other words, CAVs have the potential to reach Level 4 (high automation) and Level 5 (full automation) as defined by the Society of Automotive Engineers (SAE) [20], thanks to the use of advanced sensors and intelligent controls [21,22]. The network of these vehicles, capable of exchanging information with surrounding vehicles, road infrastructure elements, and pedestrians, is known as the Internet of Vehicles (IoV) [23].

This type of communication can occur in various ways, such as the following:Vehicle-to-Infrastructure, where the road infrastructure provides the vehicle with a stream of information, such as traffic, weather and road conditions, speed limits, and accidents. V2I is mainly concerned with the safety of the vehicle and ensures that it is able to detect dangerous situations with much greater advance.Vehicle-to-Vehicle, which allows real-time exchange of information between vehicles.Vehicle-to-Cloud, which is mainly used for downloading over-the-air vehicle updates or vehicle diagnostics or to connect with any IoT devices present in the road infrastructure.Vehicle-to-Pedestrian, which is one of the newest systems used in connected vehicles, and it is also used for safety purposes. Vehicles use sensors to detect pedestrians, which gives collision warnings.Vehicle-to-Everything, which is the combination of all the above-mentioned types of connectivity, known as V2X connectivity.

The European standard also defines the type and protocol of messages related to V2X communication, as follows:CAM (Cooperative Awareness Message) is messages that support cooperation between vehicles on the road infrastructure by indicating the status and information of each of them. The content varies depending on the device transmitting the message. In general, a CAM reports information relating to the status of the vehicle, such as position, speed, and acceleration. In addition, it reports the description of the vehicle sending the CAM, such as its size, type, and the role it plays in the infrastructure [24].DENM (Decentralized Environmental Notification Message) signals the type and location of specific events (accidents, congestion, etc.) and persists until the termination of the event [25].MAPEM (Map Extended Message) informs the ITS stations about the state of the relevant road segment. For example, a MAPEM can describe the geometry of an intersection by reporting the number of lanes, the presence of pedestrian crossings, and so on [26].SPATEM (Signal Phase and Timing Extended Message) indicates the status of one or more traffic lights at an intersection [26].SREM (Signal Request Extended Message) is used to request the modification of a traffic light following a request by an ITS station. The SSEM (Signal Status Extended Message) is the response message received by the station that previously made the request, indicating whether the request has been accepted or canceled [26].

Regarding the first scenario developed, which enables the management of an intelligent intersection, various solutions have been adopted in the literature with the following aims:Create a control system that allows the transmission and modification of SPAT and MAP, as well as the management of physical traffic lights through the use of a PLC system [27].Manage the priority of emergency vehicles through a communication system where OBUs approaching the intersection signal their presence to an RSU. The RSU, by processing the received messages, determines which vehicle should be given priority and how to adjust the traffic lights at the intersection [28].

Both solutions would be limited to intersections that are physically equipped with traffic signals. In addition, the second solution relies solely on the Road-Side Unit for managing the intelligent intersection, without taking into account the time needed to process the incoming messages. Although the second approach is effective in managing traffic lights when multiple emergency vehicles are present, the solution proposed in this paper does not consider this scenario. Instead, the objective is to notify all common vehicles approaching an intersection about the presence of one or more emergency vehicles, thereby encouraging drivers to exercise greater caution. In other words, common vehicles, each equipped with an OBU, will receive a notification regarding the presence of one or more emergency vehicles, also equipped with OBUs, when they are approaching the intersection, regardless of whether it is physically equipped with traffic lights. Notably, the notification is not managed by the RSU installed near the intersection, in order to eliminate any delay in transmitting the alert.

Regarding the second application developed to ensure the safety of those classified by ETSI as vulnerable road users (VRUs), three primary solutions are discussed in the literature aimed at warning the following:Pedestrians of an approaching vehicle through smartphone and tablet applications [29];Only the driver approaching the pedestrian crossing, encouraging greater caution for the vulnerable user in the process of crossing [27];Both the pedestrian and the vehicle driver via smartphone and tablet applications [30].

Some of these solutions detect pedestrians through additional hardware, which sends an alert to the connected vehicle using Bluetooth technology. The solutions targeting vulnerable users leverage the increasing prevalence of smartphones, particularly because a significant number of pedestrian accidents are attributed to smartphone use during crossings: of 1102 pedestrians observed, approximately 30 performed a distracting activity while crossing [31]. On the other hand, solutions aimed at warning the driver of the vehicle take into account the rising integration of infotainment systems in newer vehicles. This latter consideration was a key factor in the development of the proposed solution for the second IoT application in this paper. Moreover, studies have shown that alerting the driver of a potential accident yields better results than warning vulnerable road users [32]. This is because drivers can react more quickly to a warning than pedestrians or cyclists. Additionally, the proposed solution avoids the need to equip pedestrians with specialized hardware, as the pedestrian crossing is monitored by a traffic sensor equipped with a neural network capable of recognizing pedestrians on the crosswalk. Ultimately, the aim of the second implemented scenario is to establish communication between vehicles and the road infrastructure to indicate the presence of one or more VRUs near a crosswalk. This application is designed using an RSU connected to a traffic sensor that gathers information about the presence of VRUs on the roadway through a neural network. It then notifies nearby vehicles equipped OBUs through a mobile application.

## 3. Roadside Unit and On-Board Unit

An RSU [33] is a DSRC transceiver mounted on the sides of the road or at pedestrian crossings. The OBU [34] is an electronic device installed in a vehicle capable of collecting vehicle-related data from sensors installed on the vehicle and traffic conditions communicating with smart roads. Both devices are equipped with a DSRC transmission and reception system, a GPS receiver, a memory, and a processor capable of running ITS applications for sending inter-vehicle messages. However, while an OBU can operate in any dynamic condition (stationary and moving), an RSU can only operate if stationary. One of the most important components of such stations is the GPS receiver, which allows, in the case of OBUs, the localization of vehicles. This module has an update frequency of 10 Hz and can measure distances longer than 1.5 m. In other words, it only takes about 30 s from the activation of the ITS station to establish its geographical position. Furthermore, typically, a Road-Side Unit is connected to a cabinet that receives data and transmits them to a control center. The hardware devices used for the test scenario are manufactured by Unex Technology Corporation (official website: https://unex.com.tw/en/, acessed on 3 April 2023).

Figure 1 illustrates ITS stations responsible for running the developed IoT applications.

Each Unex Corporation device allows the development of applications related to intelligent mobility, thanks to the use of a Software Development Kit (SDK). Thanks to SDKs like the Unex one, developers can develop V2X applications very simply without necessarily having to know the transmission protocol of each message defined by the ETSI standard. In fact, it is often sufficient to simply modify configuration files to set up the protocol stack and call the APIs used for transmitting and receiving messages. The SDK, with the services made available, is loaded onto a host system and executed via the Ethernet or USB connection with the Unex device. These services include the transmission, reception, encapsulation, and decapsulation of packets; encoding and decoding of messages; and management of the security of transmitted packets. In any case, the aforementioned services are not active by default; users can activate them by uploading a configuration file in JSON format. If this is valid, it will be saved in the Unex product file system so that users do not need to load it again on the next boot. As previously stated, the SDK is installed on a host system. In this context, the host device is an IoT gateway specifically developed for smart mobility applications [35]. The combination of the RSU and the IoT gateway will be referred to as the I-RSU (Intelligent-RSU) in this paper, while the combination of the OBU and the IoT gateway will be referred to as the I-OBU (Intelligent-OBU). Additionally, both the Unex Corporation devices and the SDK are compatible with the use of Real-Time Kinematic (RTK) technology through additional modules provided by the same company. This technique increases the accuracy of GNSS positioning by utilizing correctional data obtained from a fixed base station with a known position. In the implemented scenarios, the standard GPS module provided with the devices was used, as it already delivers optimal and reliable performance.

## 4. Overall Architecture of the V2X Scenarios

The upcoming section is intended to outline the overall architecture of the established scenarios, emphasizing the essential components involved, their communication protocols, and relevant technical details.

The scenarios presented in this paper will be detailed in dedicated sections and are titled as follows:Priority Intersection Management;Pedestrian in Signalized Crosswalk Warning.

For an IoT-enabled V2X communication system like those discussed in this paper, the architecture includes the following:Terminal devices;Communication protocols between components;Data storage and processing platforms;User interfaces.

The core devices are On-Board Units installed on vehicles, Road-Side Units integrated with an infrastructure, and traffic sensors capable of detecting pedestrians. The RSUs and OBUs are physically connected to processing units via Ethernet and USB cables, respectively. Particularly, the RSU used for the “pedestrian in signalized crosswalk warning” scenario is linked to a traffic sensor through an Ethernet cable, forming a Local Area Network (LAN). Communication between OBUs and RSUs utilizes the DSRC protocol, as discussed in Section 2. Both scenarios have their processing units connected to the IoT platform ThingsBoard, which receives and processes data via the MQTT protocol. ThingsBoard provides a web-based user interface for real-time scenario monitoring. Communication with this platform is established through mobile networks, specifically 4G or 5G. The effects of network choice on communication delays with the IoT platform will be discussed in Section 5. Moreover, a mobile application sends alerts to drivers, facilitated by Bluetooth communication between the app and the processing units.

### 4.1. First Scenario—Priority Intersection Management

This scenario outlines a basic case of priority control at a single intersection for connected public transport vehicles. In an emergency situation, such as when an ambulance needs to cross an intersection quickly, the aim is to locate the emergency vehicle and notify nearby vehicles of the situation, ensuring a fast and safe passage. A mobile application specifically developed for this purpose will alert other vehicles of the emergency vehicle’s approach.

The actors involved in the exchange of V2X messages are the following:A public transport equipped with an I-OBU;A regular vehicle equipped with an I-OBU;An I-RSU.

The I-RSU dedicated to this scenario is responsible for transmitting MAPEM, which allows for forwarding a user-defined map to all nearby devices and locating vehicles within the DSRC range. In other words, this map enables vehicles equipped with I-OBUs to determine their lane and whether they need to refer to a traffic light by comparing their own coordinates with those of the defined nodes. The map transmitted by the I-RSU via MAPEM is shown in Figure 2 and represents Via Ronca in Lioni near Piazza San Carlo. Table 1 reports the coordinates of the points defined in the same figure, assuming the following:Lanes 1, 3, and 5 entering the intersection;Lanes 2 and 4 exiting the intersection.

To proceed with the automotive experiments regarding V2X communication, a system is proposed as follows:Three IoT gateways powered via a USB Type C cable from a power supply with an output of 5.1V-3A.An On-Board Unit powered via a micro USB cable from the IoT gateway. This connection also establishes communication between the IoT gateway and the OBU, mounted on the public transport vehicle.An On-Board Unit powered via a micro USB cable from the IoT gateway. This connection also establishes communication between the Raspberry and the IoT gateway, mounted on a vehicle approaching the intersection, but does not represent the emergency vehicle.An RSU powered via an Ethernet cable from a PoE power supply, which also enables V2X communication through another Ethernet cable connected to the IoT gateway.

It should be noted that the combination of the RSU or OBU with the IoT gateway is referred to as the I-RSU and I-OBU, respectively. Due to the absence of a physical traffic light, the I-RSU exclusively transmits information about the intersection’s topology to all nearby vehicles, enabling them to locate themselves at the intersection and identify their respective lanes. If the emergency vehicle is in one of the entering lanes, as indicated in the map in Figure 2, it must notify all nearby vehicles of its presence. Once the public transport vehicle has crossed the intersection, it must cease this notification. This is achieved through the emergency vehicle sending DENM and regular vehicles receiving them.

Specifically, the scenario unfolds as follows when a public transport vehicle equipped with an I-OBU enters the coverage area of an I-RSU located near the intersection. The flow of messages is outlined below:The I-RSU sends MAPEM every 100 milliseconds.The I-OBU installed in the emergency vehicle receives MAPEM from the RSU.The I-OBU, installed in the emergency vehicle, compares its position with the nodes defined in MAPEM.If the I-OBU, installed in the emergency vehicle, is located in one of the incoming lanes, it forwards DENM within the DSRC coverage area.The I-OBU on a regular vehicle receives an alert via a mobile application regarding the emergency vehicle’s presence if it is near the intersection and in one of the incoming lanes.The I-OBU of the emergency vehicle terminates the transmission of DENM after crossing the intersection.

The appearance of the designed mobile app when the alert is received is shown in Figure 3. This notification is also accompanied by an audio message indicating the proximity of an emergency vehicle.

Furthermore, specific data, such as the lanes and current positions of vehicles, are transmitted every 200 milliseconds to the IoT platform ThingsBoard using the MQTT protocol. The I-RSU sends only its operational status (on/off) to the IoT platform, while both OBUs transmit their current latitude, longitude, and lane position based on the map shown in Figure 2. Additionally, the I-OBU installed in the priority vehicle also sends a priority request alert. This configuration ensures that vehicle information and priority status can be monitored in real time by a dedicated control room. By facilitating continuous data transmission, it allows operators to track the movement and operational status of vehicles effectively. This real-time oversight not only enhances situational awareness but also allows for timely interventions when necessary, ultimately improving safety and operational efficiency within the transportation system. Figure 4 shows the ThingsBoard dashboard used for real-time monitoring. The indicators “RSU”, “Emergency Vehicle OBU”, and “OBU” display the status of each device (on/off), while the “Priority Request” indicator indicates whether the priority vehicle has submitted a priority signal to nearby vehicles. Additionally, for the regular vehicle, the lane currently occupied, as well as the previous lane, is indicated. In contrast, for the standard vehicle, only the lane currently occupied is displayed.

The overall architecture, including the various exchanged messages and the protocols used, of the developed IoT application related to the first scenario is illustrated in Figure 5.

### 4.2. Second Scenario—Pedestrian in Signalized Crosswalk Warning

A key application involves notifying intelligent vehicles of the presence of pedestrians or cyclists at pedestrian crossings, promoting safer crossings. “Blind spots” are areas outside a vehicle that the driver cannot see. The application developed for urban environments aims to improve visibility in these blind spots and alert drivers to vulnerable road users through warning signals, thereby reducing the risk of accidents. In this scenario, the objective is to transmit a warning signal via V2I communication from the infrastructure-enabled I-RSU to a vehicle equipped with an I-OBU when the vulnerable road user is present on or near the road being traversed. The alert is received by the driver of the vehicle equipped with the I-OBU through an audible and visual signal generated by a specially designed mobile application.

The actors involved in this scenario are the following:A regular vehicle equipped with an I-OBU;An I-RSU.

The I-RSU is responsible for transmitting MAPEM, which allows for the sending of a user-defined map to all nearby devices, as well as for transmitting DENM, which is necessary for forwarding collision alerts involving a VRU. As in the previous scenario, the map enables devices to determine their lane and whether they need to refer to a traffic light by comparing their coordinates with those of the defined nodes. The map transmitted by the I-RSU is visible in Figure 6 and represents another section of Via Ronca, specifically near a school called “Istituto Comprensivo Nino Iannaccone”. Table 2 presents the coordinates of the points defined in the same figure, assuming the following:Lanes 1 and 3 are entering the intersection.Lanes 2 and 4 are exiting the intersection.

To conduct vehicle-based experiments regarding V2X communication, a single vehicle near the selected intersection has been considered. The proposed system consists of the following components:An IoT gateway powered via a Type C USB cable from a power supply with an output of 5.1V-3A.An On-Board Unit powered via a micro USB cable from the IoT gateway. This connection also establishes communication between the IoT gateway and the OBU, mounted on the public transport vehicle.An RSU powered via an Ethernet cable from a PoE power supply, which also enables V2X communication through another Ethernet cable connected to the IoT gateway. This RSU is connected via a different Ethernet cable to the traffic sensor.

It is important to highlight that the integration of the RSU with the IoT gateway is designated as I-RSU, while the combination of the OBU with the IoT gateway is referred to as I-OBU. The goal of this scenario is to notify vehicles within a specific geographic area about the presence of a pedestrian on the roadway. The information regarding the presence of the pedestrian is obtained from a traffic sensor, utilizing a neural network capable of recognizing VRUs, which is connected to the I-RSU responsible for transmitting DENM. If the traffic sensor no longer detects VRUs, the I-RSU must cease message transmission. Additionally, the I-RSU must also forward MAPEM that describe the intersection topology, allowing vehicles to determine their lane position at the intersection. Vehicles should only receive notifications of the VRU’s presence in the incoming lanes, specifically the first and third lanes of the intersection shown in Figure 6.

In detail, the flow of messages exchanged between the I-OBU and the I-RSU is as follows:The I-RSU sends every 100 milliseconds MAPEM.The I-OBU establishes communication with the mobile application.The traffic sensor detects the presence of one or more vulnerable road users on the roadway.The traffic sensor transmits this information to the I-RSU.The I-RSU transmits DENM as long as the event is ongoing.The I-OBU enters the DSRC communication range and localizes itself in a lane of the intersection.The I-OBU receives a danger alert through an audible signal generated by the mobile application when it is in one of the incoming lanes at the intersection of interest.

The appearance of the app is shown in Figure 7.

In addition, a set of data is also sent to the IoT platform ThingsBoard via the MQTT protocol every 200 milliseconds. Indeed, all devices regularly transmit their operational status (on/off) to the IoT platform. The I-RSU is responsible for sending to Thigsboard an alert when the traffic sensor detects a pedestrian crossing the road. Simultaneously, the I-OBU reports its current latitude, longitude, and the lane it is occupying. This flow of information ensures real-time monitoring of both vehicle location and pedestrian presence. The ThingsBoard dashboard, as illustrated in Figure 8, includes a map that shows the real-time locations of vehicles. Additionally, the “RSU” and “OBU” indicators represent the operational status (on/off) of each respective device, while the “Pedestrian” indicator signals the presence of a vulnerable road user in the process of crossing. Moreover, a colored circle on the map outlines the DSRC coverage area within which notifications are transmitted, corresponding to the lane currently occupied. The circle’s color (red/green) adjusts based on the status of the “Pedestrian” indicator: it turns red when active, and green when inactive. The map also provides a visual indication of the lane occupied by each vehicle.

The overall architecture, including the various exchanged messages and the protocols used, of the developed IoT application related to the second scenario is visible in Figure 9.

## 5. Latencies Between Devices and the IoT Platform

V2I communication between smart roads and intelligent vehicles is essential for building safe roads and also for more effective traffic management. This section addresses the latency experienced between various devices and the IoT platform ThingsBoard, detailing the response times and factors affecting communication efficiency. Understanding these latencies is crucial for optimizing system performance and ensuring timely data transmission, which is essential for effective real-time monitoring and communication in IoT-enabled V2X scenarios. Since the messages sent by the RSUs occur on an as-needed basis, these devices are not taken into account: the operational status of the RSUs is reported only during device activation and deactivation, while pedestrian alerts are transmitted solely when the relevant event occurs. Consequently, tests were conducted to assess the latencies in V2I communication between vehicles and the IoT data acquisition platform, considering that telemetry data from the vehicle are transmitted every 200 ms. In particular, the focus is on the time discrepancy between two successive acquisitions on the IoT platform, utilizing both 4G and 5G data networks.

Figure 10 illustrates the results obtained using the LTE network. The latency values, compared with the configured transmission rate of 200 ms to the IoT platform via the OBUs, reveal numerous peaks exceeding 400 ms, indicating that the 4G network is insufficient for managing communications with remote infrastructures at this cycle time. To mitigate these latencies, a cycle time greater than 800 ms would be necessary, as this represents the highest observed peak. For instance, at a speed of 50 km/h, a vehicle would travel approximately 11 m between transmissions with an 800 ms cycle, compared with 2.7 m at 200 ms, leading to significant information loss, especially regarding smart road interactions, such as pedestrian crossings and priority intersections. Conversely, a 200 ms cycle with LTE tends to fill the buffers managed by the IoT platform more quickly, as it runs on a server also handling other tasks. Consequently, latencies below 200 ms indicate substantial queuing on the remote platform’s buffer, caused by prior peaks well above the cycle time. The trends reveal that the highest peaks correspond to subsequent samples with latencies under 200 ms. To efficiently process these queues and quickly clear the accumulated packets from the buffers, the platform writes telemetry data sequentially at intervals below 200 ms. This signifies a significant temporal misalignment between the packets transmitted by vehicles and the database writing.

This misalignment could be addressed by increasing the telemetry transmission cycle time (e.g., at least 800 ms) or utilizing a faster data network, such as 5G, capable of handling shorter cycle times. As a demonstrative conclusion, Figure 11 shows only the latencies related to the “pedestrian in signalized crosswalk warning” scenario using the 5G data network. It is observed that these latencies are significantly lower compared with the analogous case with LTE communication (Figure 10), and the trends exhibit a more regular pattern around 200 ms. Thus, while the available bandwidth with LTE causes delays that accumulate and generate significant queuing on the IoT platform, when greater bandwidth is utilized, the delays between telemetry transmission from the vehicle and ingestion on the platform are reduced. The existing channel struggles to manage the volume of telemetry data accumulating every 200 ms due to its limited bandwidth. In contrast, when the channel offers greater bandwidth, it facilitates smoother packet transmission with reduced queuing. Consequently, the platform can process queues more quickly, significantly reducing the temporal misalignment between the vehicle and the infrastructure. Therefore, in this case, under the worst conditions, to minimize latencies, the cycle time could be set to 500 ms, which represents a good trade-off between the initially hypothesized 200 ms and the 800 ms required with an LTE data network.

## 6. Discussion

The functionality of the scenarios presented in this paper mainly depends on the following two factors:The quality of the signal received from the GPS module;The scalability of the DSRC protocol in congested environments.

In addressing the first issue, it is essential to note that for the GNSS module to provide accurate positioning, it must receive signals from at least four satellites [36]. Consequently, the more satellites from which the ITS station module receives a signal, the better the localization quality will be. This means that a poor localization quality of the station could lead the vehicle to position itself in a different lane on the map than the one it is actually in. This issue could result in a failure to send or receive the alert signal and worsens as the width of the roadway decreases. Ideally, the GPS antenna should be placed in an environment free of buildings and where the sky is completely unobstructed, as buildings can block the GPS signal, preventing the receiver from receiving signals from certain satellites [37]. This condition is difficult to meet, but the problem can be mitigated by meeting minimum positioning requirements. One of these requirements is positioning the GPS receiver outside the vehicle. It has been observed that placing the GPS module outside the vehicle significantly improves signal reception, ensuring reliable localization in most cases. However, for situations where signal reception cannot be guaranteed, failover mechanisms have been implemented. These mechanisms include periodic restarts of the software system until a GPS signal is successfully acquired. To address cases where the vehicle cannot achieve precise localization due to GPS module errors, an additional logic has been implemented. This logic evaluates the vehicle’s coordinates to determine whether it is approaching or moving away from the intersection of interest. In scenarios where the vehicle cannot accurately localize itself in the correct lane or is mistakenly associated with a different lane, this control logic ensures that all vehicles approaching the intersection still receive the alert signal.

Regarding the scalability of the DSRC protocol, various studies emphasize its advantages and challenges in congested environments. The DSRC protocol is frequently recognized for its low-latency communication capabilities, rendering it suitable for numerous V2X applications. Specifically, it is noted for maintaining strong data transfer performance, even under challenging conditions, such as physical obstacles or severe weather. This reliability positions DSRC as a crucial foundation for V2X applications, applicable in both densely populated urban areas and rural settings. However, it has scalability challenges, particularly in high-density traffic, where an increased number of vehicles can lead to packet collisions and reduced performance. One study benchmarks DSRC against LTE-based Cellular V2X (C-V2X) and shows that while DSRC performs better with lower latency and higher packet delivery ratios in congested conditions, it suffers from limitations as vehicle density rises, especially when exceeding 1000 vehicles per hour [38,39]. Several studies suggest congestion control mechanisms, such as adjusting the data rate or transmission power, to mitigate these issues. For example, decentralized congestion control algorithms that dynamically adjust the data rate based on channel load have shown promise in improving the performance of DSRC in VANETs [40,41]. Alternative technologies to the DSRC protocol, such as C-V2X, were not considered due to their shared use of the same frequency band as DSRC, which subjects them to similar congestion issues under high data traffic conditions [42].

Another important consideration is the latency in communication with the IoT platform, along with ensuring the security of data transmitted between vehicles and the platform. In relation to the first issue, it is important to note that the platform was designed as a central control hub to assist municipalities in managing traffic, as well as coordinating emergency services and safety interventions. High latency in this context can have significant consequences on the system’s performance. Delayed updates can reduce the effectiveness of real-time monitoring systems, which are crucial for making immediate decisions in traffic management and emergency response. Furthermore, delays in data transmission can lead to inconsistencies between devices and the central platform, complicating the synchronization of historical records. It is important to specify that the study of latencies between the data sent by vehicles to the IoT platform does not affect the functionality of the two scenarios but only the real-time remote monitoring. In other words, the notification of an approaching emergency vehicle or one or more pedestrians on the road will always be received via the DSRC protocol by the equipped vehicles, while remote users may occasionally experience delays in data observation from the IoT platform.

The security of communications between vehicles and a road infrastructure is paramount to ensure the safety and reliability of intelligent transportation systems. Any vulnerability in these communications could expose vehicles to cyberattacks, leading to potentially dangerous situations. For example, if an attacker gains access to the communication network, they could send false traffic signals to a vehicle, causing it to make dangerous driving decisions, such as failing to stop at a red light or improperly reacting to an emergency vehicle. Ensuring secure communication channels between vehicles and an infrastructure, such as traffic lights or road sensors, through robust encryption and authentication mechanisms, is crucial to prevent such threats. The security of V2X messages is ensured through a combination of advanced encryption techniques, secure authentication mechanisms, and integrity checks. The security of V2X messages is ensured through a comprehensive authentication mechanism that relies on certificates to verify the sender’s identity and permissions. These certificates, as outlined in [43], authorize the transmission of specific message types between ITS-Ss (Intelligent Transport Systems stations). The certificate includes two key identifiers: the ITS-Application Identifier (ITS-AID) and the Service Specific Permissions (SSPs). ITS-AID, as defined in [44], indicates the type of message the sender is allowed to transmit. SSP further refines the scope by specifying particular permissions within the broader ITS-AID, for example, in the case of a CAM, allowing the sender to transmit messages based on the vehicle role specified within the message itself. This structure ensures that only authorized entities can send V2X messages, protecting the system from unauthorized or malicious messages. As a result, each incoming signed message is carefully validated, confirming both the authenticity of the message and the legitimacy of its sender before being accepted into the system. Regarding the security of communication with the IoT platform, it is essential to highlight that it is managed through the MQTT protocol, which integrates several key security features. Indeed, data transmission occurs by encrypting the data and ensuring its integrity through hashing algorithms. In addition, the protocol supports authentication mechanisms, including username/password combinations and client certificates, which verify the identity of both devices and users. This allows the platform administrator to define specific permissions, ensuring that only authorized entities can publish or subscribe to designated topics.

## 7. Conclusions and Future Works

The two V2X communication scenarios tested show great potential in improving both road safety and traffic management. The implementation of a system that allows emergency vehicles to pass through intersections efficiently, and another that ensures the safe crossing of vulnerable road users, highlights the capability of V2I technologies to create a more responsive and secure transportation ecosystem. The solution developed for emergency vehicles proves to be particularly effective in densely populated urban areas, significantly reducing accidents and improving traffic flow. In contrast, the solution for protecting vulnerable road users is especially effective in situations where a vehicle does not have a clear line of sight to a pedestrian, such as before a turn. By leveraging real-time communication and IoT platforms, these scenarios demonstrate the practical benefits of V2X technologies for smarter, safer cities. However, there are certain limitations to consider, particularly regarding scalability, as a high number of vehicles typical in large metropolitan areas could affect the quality of V2X communications. In these scenarios, alerts are broadcast to all vehicles within the DSRC communication range. However, in larger urban areas, these alerts could reach dozens of vehicles, necessitating careful consideration of scalability in such systems. Therefore, these scenarios need to be adjusted to accommodate the higher volume of traffic typically seen in metropolitan areas. The integration of traffic light control makes these solutions particularly suitable for large cities. It is important to note that, while the intersections selected do not feature traffic lights, both scenarios can be easily adapted to traffic light control. In the traffic management scenario, MAPEM can be seamlessly integrated with SPATEM, SREM, and SSEM to manage traffic lights effectively. For instance, when the section of the road traversed by the emergency vehicle has a red light, it can send a priority request to the I-RSU through an SREM. The I-RSU will then transmit a response message indicating whether the request has been accepted or denied. If accepted, the SPATEMs will be modified to turn the traffic light green for the emergency vehicle’s route, while setting all other lights to red. Through this straightforward extension of the implemented project, the emergency vehicle can take advantage of a series of synchronized green lights to ensure smooth passage through intersections. A similar strategy is employed for the scenario concerning vulnerable road users. When a pedestrian is using the crosswalk, the I-RSU equipped with a traffic sensor can modify the traffic light’s status until the pedestrian has safely crossed, thereby ensuring the safety of vulnerable road users at the intersection. This article has demonstrated how these solutions have been successfully implemented in a smaller town like Lioni, showcasing their practical application and effectiveness in real-world scenarios. With appropriate adjustments to scalability and infrastructure, these V2X communication solutions can be seamlessly integrated into more complex, high-traffic environments, ensuring continued benefits in road safety and traffic management and the protection of vulnerable road users.

## Figures and Tables

**Figure 1 sensors-25-00485-f001:**
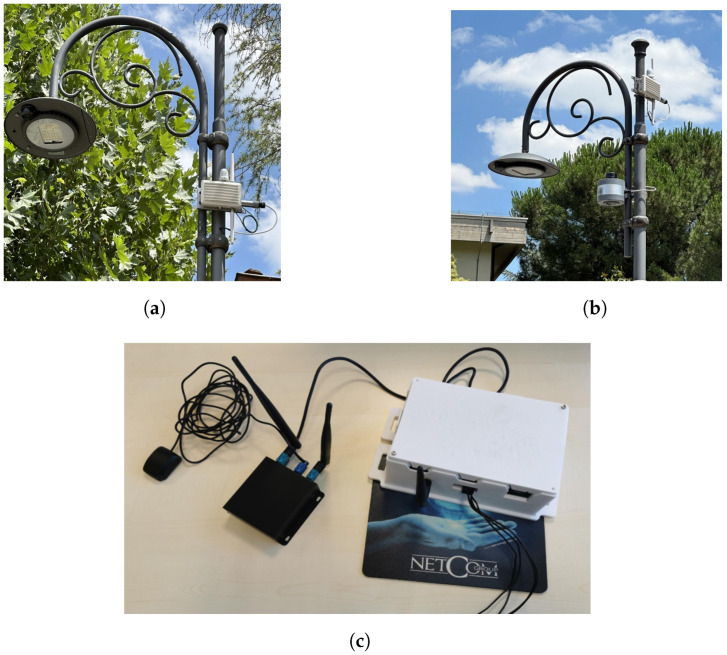
ITS stations used for the implementation of designed V2I communication scenarios. (**a**) Road-Side Unit installed on Via Ronca in Lioni (Campania, Italy) on a public lighting element for the scenario dedicated to the priority passage of an emergency vehicle. (**b**) Road-Side Unit and traffic sensor installed on Via Ronca in Lioni (Campania, Italy) on a public lighting element for the scenario dedicated to ensuring the safety of VRUs. (**c**) System composed of an On-Board Unit and IoT gateway, utilized in scenarios dedicated to the management of intersection priority and the safety of VRUs.

**Figure 2 sensors-25-00485-f002:**
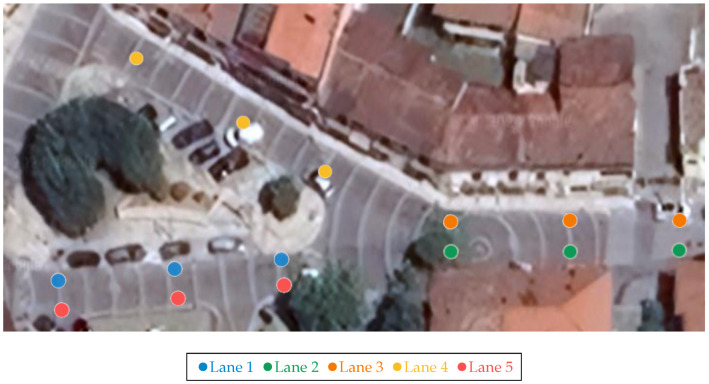
Satellite map of the intersection on Via Ronca dedicated to the priority intersection management, with the definition of nodes and lanes transmitted via the MAPEMs from the RSU.

**Figure 3 sensors-25-00485-f003:**
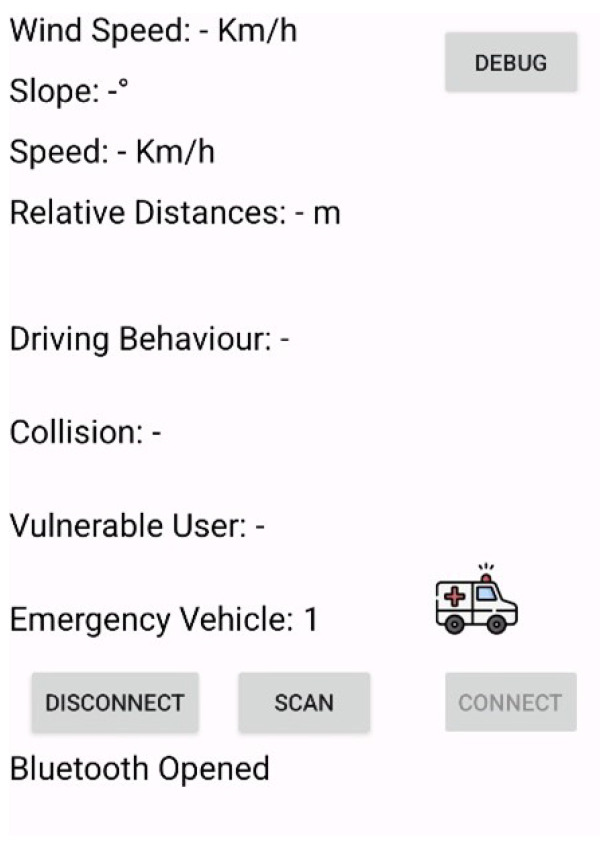
Appearance of the mobile app upon receiving the notification sent by the emergency vehicle for the “priority intersection management” scenario.

**Figure 4 sensors-25-00485-f004:**
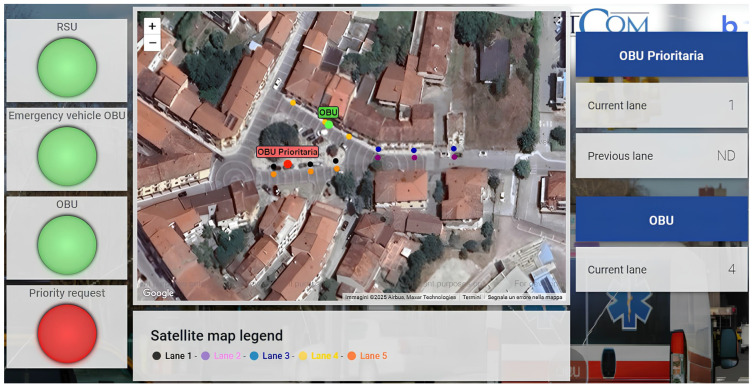
Real-time ThingsBoard dashboard for the “priority intersection management” scenario.

**Figure 5 sensors-25-00485-f005:**
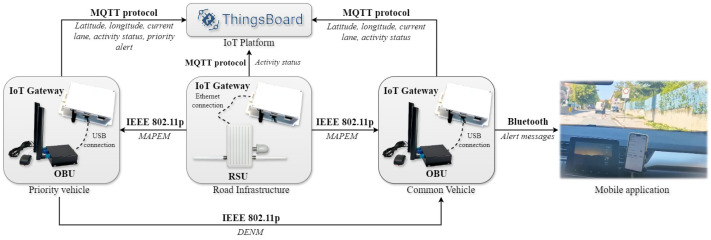
Overall architecture, exchanged messages, and protocols used in the “priority intersection management” scenario.

**Figure 6 sensors-25-00485-f006:**
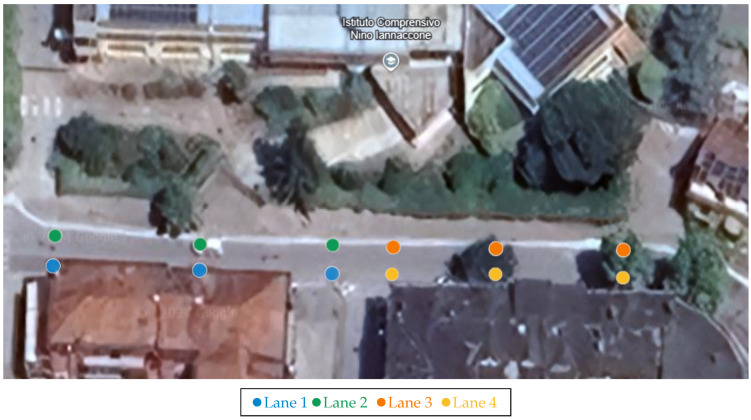
Satellite map of the intersection on Via Ronca dedicated to the “pedestrian in signalized crosswalk warning” scenario, with the definition of nodes and lanes transmitted via MAPEM from the RSU.

**Figure 7 sensors-25-00485-f007:**
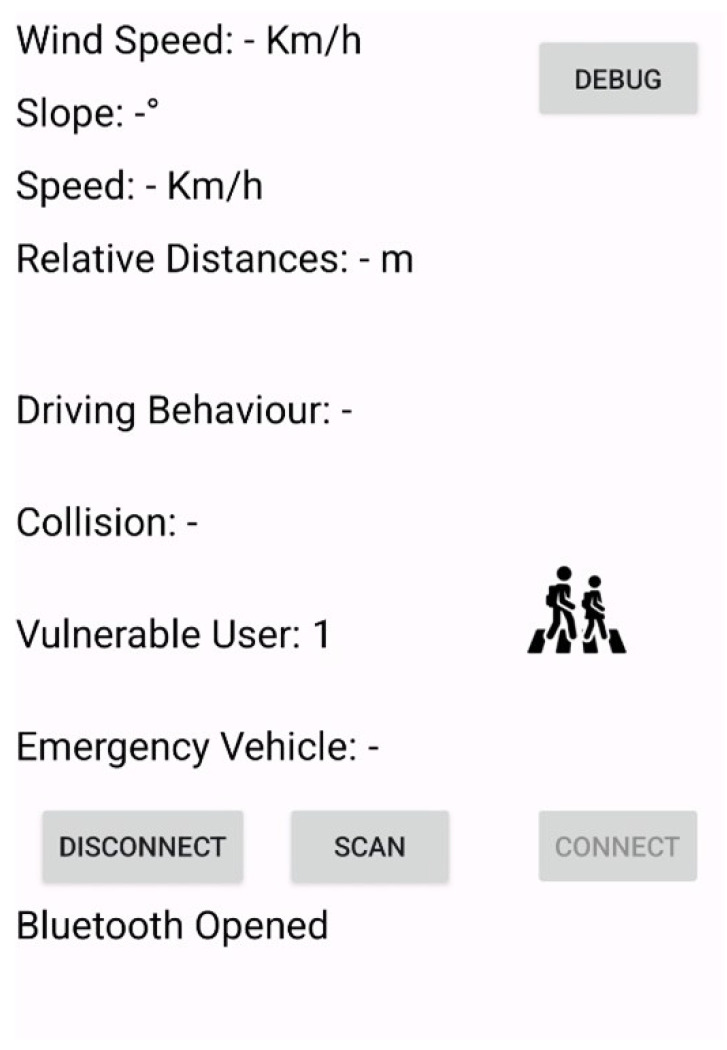
Appearance of the mobile app upon receiving the notification sent by RSU for the “pedestrian in signalized crosswalk warning” scenario.

**Figure 8 sensors-25-00485-f008:**
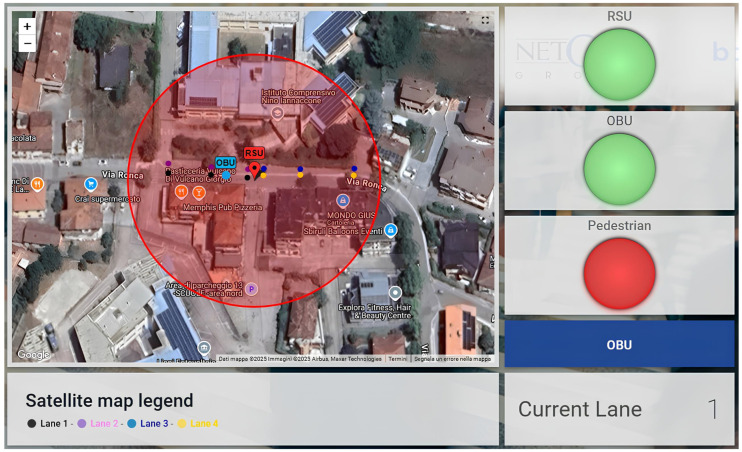
Real-time ThingsBoard dashboard for the “pedestrian in signalized crosswalk warning” scenario.

**Figure 9 sensors-25-00485-f009:**
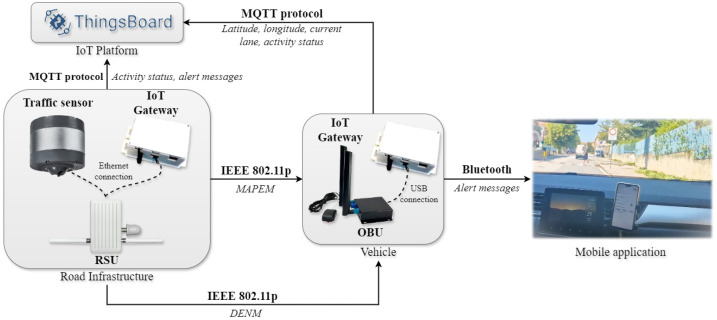
Overall architecture, exchanged messages, and protocols used in the “pedestrian in signalized crosswalk warning” scenario.

**Figure 10 sensors-25-00485-f010:**
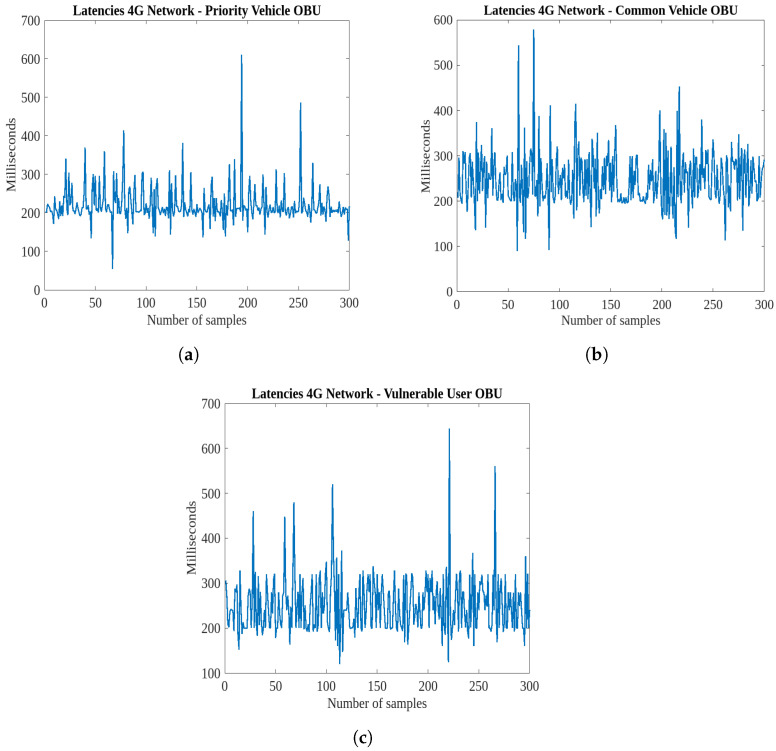
Latencies in V2I communication between the vehicles and the IoT platform, utilizing a 4G data network. (**a**) Latency between the OBU on the priority vehicle and the IoT platform in the “priority intersection management” scenario. (**b**) Latency between the OBU on the common vehicle and the IoT Platform in the “priority intersection management” scenario. (**c**) Latency between the OBU on the vehicle and the IoT Platform in the “pedestrian in signalized crosswalk warning” scenario.

**Figure 11 sensors-25-00485-f011:**
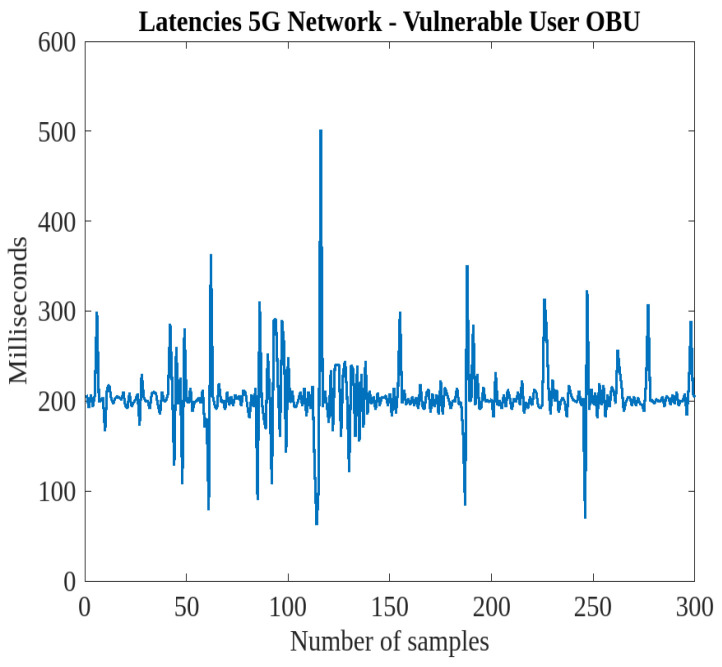
Latencies in V2I communication between the vehicle and the IoT platform in the “pedestrian in signalized crosswalk warning” scenario, utilizing a 5G data network.

**Table 1 sensors-25-00485-t001:** Definition of nodes and lanes on the map transmitted by MAPEMs.

Node (Latitude, Longitude)	Lane 1 (Incoming)	Lane 2 (Outgoing)	Lane 3 (Incoming)	Lane 4 (Outgoing)	Lane 5 (Outgoing)
1	40.8763198, 15.1898582	40.8763323, 15.1900638	40.8763624, 15.1900654	40.8764092, 15.1899234	40.8762917, 15.1898636
2	40.8763077, 15.1897352	40.8763316, 15.1902412	40.8763592, 15.1902367	40.8764619, 15.1898079	40.8762811, 15.1897385
3	40.8762994, 15.1895583	40.8763350, 15.1904321	40.8763643, 15.1904312	40.8765329, 15.1896538	40.8762720, 15.1895595

**Table 2 sensors-25-00485-t002:** Definition of nodes and lanes on the map transmitted by MAPEMs.

Node (Latitude, Longitude)	Lane 1 (Incoming)	Lane 2 (Outgoing)	Lane 3 (Incoming)	Lane 4 (Outgoing)
1	40.8762912, 15.1912553	40.8763227, 15.1912570	40.8763241, 15.1913323	40.8762994, 15.1913297
2	40.8763011, 15.1910812	40.8763295, 15.1910837	40.8763228, 15.1915040	40.8763012, 15.1915033
3	40.8763084, 15.1908801	40.8763413, 15.1908816	40.8763234, 15.1917580	40.8762987, 15.1917553

## Data Availability

The data used in this research are stored in private corporate repositories and are therefore not publicly available. Additionally, the SDK required for implementing the scenarios is provided by Unex Corporation upon purchasing their products. However, we are open to sharing any available data upon request from interested researchers.

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
