# Peer review of "Implementation and Testing of V2I Communication Strategies for Emergency Vehicle Priority and Pedestrian Safety in Urban Environments"

_sensors, 2025, doi:10.3390/s25020485_

Round 1

Reviewer 1 Report

Comments and Suggestions for Authors

Journal: Implementation and Testing of V2I Communication Strategies for Emergency Vehicles Priority and Pedestrian Safety in Urban Environments

Manuscript ID: sensors-3349724

I hereby confirm the acceptance of the paper, pending minor revisions. The abstract effectively outlines the key objectives and scenarios tested, focusing on the impact of Vehicle-to-Infrastructure (V2I) communication in improving road safety for emergency vehicles and pedestrians. The results, which demonstrate significant improvements in response times and pedestrian safety, highlight the important contribution of this research to advancing intelligent transportation systems and creating safer urban environments. This study is important both in terms of facilitating the passage of emergency vehicles and contributing to pedestrian safety. I recommend that the authors review the study titled "Pedestrian safety at signalized intersections: Spatial and machine learning approaches." This work may provide valuable insights and methodologies that could enhance the current research, particularly in relation to pedestrian safety at intersections.

The literature gap should be more clearly explained. Providing a detailed analysis of the existing research and identifying the specific areas that remain underexplored or insufficiently addressed will help highlight the novelty and significance of the current study. This would offer readers a clearer understanding of the research's contribution to filling those gaps.

The abstract is currently quite general and lacks specific details about the work and its outcomes. It would be beneficial to revise it to better reflect the study's objectives, methodology, and key results. Adding more specifics about the scenarios tested, the technology used, and the precise findings (such as the exact reductions in response times for emergency vehicles or the improvements in pedestrian safety) would make the abstract more informative and reflective of the study’s contributions. This revision would provide readers with a clearer understanding of the work's significance and impact on intelligent transportation systems.

The high speeds at which ambulances travel to respond to emergencies contribute 29 significantly to the frequency of these accidents [6]. To make the reference numbering start from 1 instead of 6, you would need to adjust the citation style or reference list in the document itself. The specific method depends on the citation style you're using (APA, MLA, IEEE, etc.).

Current data should be researched; if unavailable, the provided information should be referenced. “In 2022, a total of 224 people lost their lives in crashes involving emergency vehicles in the United States. Of these fatalities, 50% were occupants of non-emergency vehicles, while 22% were pedestrians. Moreover, ambulance accidents can pose serious risks, as they often involve patients who are already injured or unwell.”

In December 2023, the National Fire Protection Association (NFPA) any update for 2024?

Reviewer 2 Report

Comments and Suggestions for Authors

This paper presents an interesting and relevant contribution to the field of V2I communication, particularly with its focus on emergency vehicle prioritization and pedestrian safety. However, there are some fragilities in the discussion that should be addressed to enhance the robustness and applicability of the research.

Specifically, I would recommend that the authors provide detailed responses to the following questions:

  1. How do you plan to ensure reliable GPS localization in urban environments with obstructed views or narrow streets?
  2. Have you considered integrating alternative localization technologies (e.g., VLC, RTK, SLAM) to enhance positioning accuracy?
  3. What specific congestion control mechanisms were tested or proposed for mitigating DSRC's scalability issues in high-density traffic?
  4. Have you considered alternative communication protocols or hybrid systems, such as Cellular-V2X, to address scalability challenges?
  5. Were worst-case scenarios (e.g., simultaneous emergencies or heavy interference) tested? If not, how do you anticipate handling such conditions?
  6. How representative are the testing conditions of larger urban environments, and how do you plan to adapt the system for more complex scenarios?
  7. What additional redundancy measures or fallback mechanisms are in place to handle communication failures or GPS signal loss?
  8. What are the implications of IoT platform latency for long-term monitoring, data analytics, or integration with smart city systems?
  9. What measures are in place to address data privacy and security concerns associated with real-time communication and IoT usage?

By addressing these points, the paper can provide a stronger and more comprehensive discussion, ensuring its relevance and scalability for real-world applications.
